# Performance, Reaction Pathway and Kinetics of the Enhanced Dechlorination Degradation of 2,4-Dichlorophenol by Fe/Ni Nanoparticles Supported on Attapulgite Disaggregated by a Ball Milling–Freezing Process

**DOI:** 10.3390/ma15113957

**Published:** 2022-06-02

**Authors:** Hongdan Wu, Junwen Wang, Hong Liu, Xianyuan Fan

**Affiliations:** 1College of Resource and Environmental Engineering, Wuhan University of Science and Technology, Wuhan 430081, China; wu_dan725@wust.edu.cn (H.W.); wangjunwen@wust.edu.cn (J.W.); fanxianyuan@wust.edu.cn (X.F.); 2Hubei Key Laboratory for Efficient Utilization and Agglomeration of Metallurgic Mineral Resources, Wuhan University of Science and Technology, Wuhan 430081, China

**Keywords:** disaggregated attapulgite, Fe/Ni bimetallic nanoparticles, 2,4-dichlorophenol, dechlorination degradation, reaction pathway, kinetics

## Abstract

Attapulgite (ATP) disaggregated by a ball milling–freezing process was used to support Fe/Ni bimetallic nanoparticles (nFe/Ni) to obtain a composite material of D-ATP-nFe/Ni for the dechlorination degradation of 2,4-dichlorophenol (2,4-DCP), thus improving the problem of agglomeration and oxidation passivation of nanoscale zero-valent iron (nFe) in the dechlorination degradation of chlorinated organic compounds. The results show that Fe/Ni nanoparticle clusters were dispersed into single spherical particles by the ball milling–freezing-disaggregated attapulgite, in which the average particle size decreased from 423.94 nm to 54.51 nm, and the specific surface area of D-ATP-nFe /Ni (97.10 m^2^/g) was 6.9 times greater than that of nFe/Ni (14.15 m^2^/g). Therefore, the degradation rate of 2,4-DCP increased from 81.9% during ATP-nFe/Ni application to 96.8% during D-ATP-nFe/Ni application within 120 min, and the yield of phenol increased from 57.2% to 86.1%. Meanwhile, the reaction rate *K*_obs_ of the degradation of 2,4-DCP by D-ATP-nFe/Ni was 0.0277 min^−1^, which was higher than that of ATP-nFe/Ni (0.0135 min^−1^). In the dechlorination process of 2,4-DCP by D-ATP-nFe/Ni, the reaction rate for the direct dechlorination of 2,4-DCP of phenol (*k*_5_ = 0.0156 min^−1^) was much higher than that of 4-chlorophenol (4-CP, *k*_2_ = 0.0052 min^−1^) and 2-chlorophenol (2-CP, *k*_1_ = 0.0070 min^−1^), which suggests that the main dechlorination degradation pathway for the removal of 2,4-DCP by D-ATP-nFe/Ni was directly reduced to phenol by the removal of two chlorine atoms. In the secondary pathway, the removal of one chlorine atom from 2,4-DCP to generate 2-CP or 4-CP as intermediate was the rate controlling step. The final dechlorination product (phenol) was obtained when the dechlorination rate accelerated with the progress of the reaction. This study contributes to the broad topic of organic pollutant treatment by the application of clay minerals.

## 1. Introduction

2,4-dichlorophenol (2,4-DCP), a typical refractory organic pollutant, is widely used in the industrial production of pesticides, disinfectants, dyes and leather [1,2,3]. It is well known that the self-condensation reaction of phenolic substances produces dioxins. Additionally, 2,4-DCP has a strong resistance to degradation due to its aromatic ring and chlorinated atoms, which make it difficult to meet the requirements of emission standards by conventional processes and biological treatments [4,5].

The application of zero-valent iron (Fe^0^) in the dechlorination degradation of chlorinated organic compounds had been extensively explored in recent years due to its low cost, high activity and non-toxicity [6,7]. Compared to micron-sized particles, Fe^0^ with particle size at nanoscale (nFe) exhibited more reactive activity due to its larger surface area [8,9,10,11]. However, nFe had some disadvantages in the application of water pollution control and in situ remediation, such as rapid agglomeration and easy oxidation passivation, which can affect subsequent dechlorination degradation [12,13,14]. To improve this issue, porous materials, such as bentonite, montmorillonite and activated carbon, have been employed as a support by previous researchers [15,16,17], in which the agglomeration of nanoparticles could be prevented by the steric hindrance of porous materials. Meanwhile, the porosity of the support could accelerate the transfer of pollutants to the reactive site, thus promoting the degradation of pollutants. Additionally, as the chlorine atoms in 2,4-DCP were connected with benzene and difficult to be removed, bimetallic nanoparticles, such as Fe/Ni, Fe/Pd and Fe/Cu, were obtained by depositing a secondary metal on the surface of nFe that could improve the dechlorination efficiency of nFe [18,19,20].

Attapulgite (ATP), which is a kind of natural nanomaterial with a fibrous rod-like microstructure (0.5–5.0 μm in length and 20–70 nm in diameter), has a potential application in contaminated groundwater remediation and wastewater treatment owing to its advantageous properties, such as wide source, large specific surface area, relatively low cost and good absorption [21,22,23]. However, the fibrous crystals of natural attapulgite prefer to exist as aggregates or crystal bundles due to the interparticle Van der Waals forces and hydrogen bonding interactions [24], which make it difficult to give full play to its nanometer characteristics and large specific surface area. Hence, the high-efficient disaggregation of attapulgite clusters into smaller crystal bundles/single crystals is the key to utilize its nanometer properties and develop the related functional materials [25]. To date, physical methods have been employed to disperse attapulgite aggregates. Dry processes, such as extrusion, ball milling and grinding, can destroy the microstructure of attapulgite, whereas wet processes, including colloid grinding, high-speed beating and ultrasonication, present the problems of hard separation for a liquid–solid system and high-energy consumption ultrasonication [26].

In this work, attapulgite clusters are disaggregated to nanoscale crystal rods by a semi-dry process due to the hydrophilic porous silicate properties of attapulgite and the volumetric expansion characteristics of the water-phase transition, in which the freezing process was combined with a ball-milling treatment. Thereafter, the disaggregated attapulgite (D-ATP) was used as a support of Fe/Ni bimetallic nanoparticles to form a composite material of D-ATP-nFe/Ni, which was applied to the dechlorination degradation of 2,4-DCP. On one hand, the disaggregated attapulgite as a support had a smaller size and larger specific surface area compared with bentonite, montmorillonite, activated carbon and other porous solid materials, thus playing a better dispersion role and avoiding the agglomeration of nFe particles. On the other hand, the introduction of Ni could catalyze the dissociation of H_2_ to active hydrogen atom (H*) by the reaction between Fe^0^ and water, which could improve the efficiency of dechlorination by breaking the C-Cl bond on the benzene ring. Meanwhile, the deposition of Ni on the surface of nFe particles could reduce the oxidization of Fe^0^. The materials were characterized by TEM, XRD and BET, and the enhancement effect of dechlorination performance of 2,4-DCP by D-ATP-nFe/Ni was investigated. The reaction pathway and kinetics of dechlorination of 2,4-DCP by D-ATP-NFe/Ni were conducted by analyzing the concentration change and conversion rate of the degradation products.

## 2. Experimental Procedure

### 2.1. Materials and Chemicals

Attapulgite (ATP) was obtained from Mingmei Mineral Co. Ltd. (Anhui, China). Chemicals including nickel sulphate (NiSO_4_·6H_2_O), ferrous sulfate (FeSO_4_·7H_2_O), sodium borohydride (NaBH_4_) and 2,4-dichlorophenol (C_6_H_4_Cl_2_O, 2,4-DCP) were of analytical grade and were purchased from Sinopharm Chemical Reagent Co. Ltd (Shanghai, China).

### 2.2. Preparation of Materials

#### 2.2.1. Preparation of D-ATP

Initially, 5 g of dried ATP was soaked in 40 mL deionized water for 2 h, before being ball-milled for 10 min in a planetary ball mill at 650 r/min. The as-prepared material was dried under vacuum in a Freeze Dryer at −20 ℃ for 20 h. The dried material was ground to powder to generate the D-ATP.

#### 2.2.2. Preparation of D-ATP-nFe/Ni

First, 2.24 g of D-ATP and 20 mL of 1 M FeSO_4_ solution were put into a three-necked flask in a nitrogen (N_2_) atmosphere. After the mixture was mechanically stirred for 30 min, 40 mL of 1 M NaBH_4_ solution was dropwise added to the mixture and then the mixture was further stirred for 15 min. Subsequently, 0.39 mL of 1 M NiSO_4_ solution was added to the mixture (the mass ratio of Ni to Fe was 0.02). After reacting for 20 min, the prepared material was vacuum-filtered and washed with deionized water 3 times, and dried under vacuum in a Freeze Dryer for 14 h; thus, D-ATP-nFe/Ni was obtained.

The same procedure was used to obtain the ATP-nFe/Ni composite material and Fe/Ni bimetallic nanoparticles (nFe/Ni), and nFe/Ni was prepared under identical conditions without adding ATP or D-ATP. All materials were freshly prepared before the experiment to avoid oxidation.

### 2.3. Characterizations of Materials

The morphologies of the materials were observed by Transmission Electron Microscope (FEI Tecnai G2 F30, Thermo Fisher, Waltham, MA, USA). The XRD patterns of the samples were recorded on an X-ray power diffractometer (D/MAX-2500, Rigaku, Tokyo, Japan) with Cu-Kα radiation operated at 40 kV and 40 mA. Specific surface area, micropore and mesoporous volume as well as pore size were determined at 77 K on a static volumetric sorption analyzer (ASAP 2020, Micromeritics, Norcross, GA, USA). The redox potential (ORP or Eh) of deionized water, FeSO_4_ solution, 2,4-DCP solution and synthesized materials were measured using an electrochemical workstation (CHI-660E, CH Instruments, Inc., Austin, TX, USA). The individual Eh measurements lasted for 65 min, and their corresponding Eh values (as determined by the electrochemical workstation) were analyzed. The content of Fe in the material was measured by a device (FH-1, INESA Co., Shanghai, China), which was based on the reaction of Fe with diluted hydrochloric acid. The content of the Fe could be determined by measuring the H_2_ generated by the reaction. The concentration of dissolved H_2_ was measured using a portable dissolved hydrogen meter (ENH 1000, Trustlex, Okayama, Japan).

### 2.4. Degradation Experiments of 2,4-DCP

Firstly, 0.4 g material (ATP, D-ATP, nFe/Ni, ATP-nFe/Ni, and D-ATP-nFe/Ni, respectively) was added to a 200 mL 2,4-DCP solution (10.0 mg/L) with a pH of 5.5–6.0 (the dosage was 2 g/L). The mixed solution was vibrated at 200 rpm under 25 °C on a thermostatic oscillator. Subsequently, the mixture was collected at determined time intervals and filtered through a 0.45 μm membrane to determine the concentration of 2,4-DCP and byproducts. The concentrations of 2,4-DCP, 2-chlorophenol (2-CP), 4-chlorophenol (4-CP) and phenol (PN) in the solution were determined by high-performance liquid chromatography (HPLC, Thermo Fei UltiMate 3000, Thermo Fisher, Dreieich, Germany).

## 3. Results and Discussion

### 3.1. Structure and Physicochemical Properties of Materials

#### 3.1.1. TEM Analyses

The TEM images of the five materials (ATP, D-ATP, nFe/Ni, ATP-nFe/Ni and D-ATP-nFe/Ni) are shown in Figure 1. The unprocessed ATP shows large crystal bundles or single needle-like crystals (Figure 1a). The ball-milled frozen ATP (D-ATP) was effectively disaggregated, and a number of dispersed single ATP crystal rods with a length of 0.5–1.5 μm and a diameter of 20–40 nm could be observed (Figure 1b), thus indicating that the D-ATP was well disaggregated without changing the length-diameter ratio of crystal rods during the disaggregation process (Figure 1b). This was due to the fact that the extrusion, shear and circumferential stresses generated during the ball-milling process could produce a large shear effect on ATP, thus promoting the disaggregation of crystal bundles. Meanwhile, the volumetric expansion of water in ATP crystal led to the disaggregation of crystal bundles in the freezing process [27,28]. Figure 1c showed that nFe/Ni particles connected each other to form a chain-like structure due to magnetic force, and some clusters formed larger agglomeration. nFe/Ni clusters were well dispersed as nFe/Ni particles loaded on ATP (Figure 1d) mainly because the steric hindrance of ATP effectively inhibited the formation of nFe/Ni clusters. Moreover, as shown in Figure 1e, the agglomeration of nFe/Ni particles was significantly improved when D-ATP was employed as the support; a few nFe/Ni clusters were even dispersed into lots of single nFe/Ni spherical particles with smaller size, indicating that D-ATP could play a better dispersion effect.

#### 3.1.2. XRD Analyses

The powder diffraction patterns of the materials (ATP, D-ATP, nFe/Ni, ATP-nFe/Ni and D-ATP-nFe/Ni) are presented in Figure 2. The diffraction peaks of ATP before and after disaggregation were all at the diffraction angles of 16.5°, 21.4°, 24.1° and 35.8°, which were attributed to the characteristic peaks of attapulgite (JCPDS 20-0688). This suggests that the disaggregated attapulgite retained its original structure without any damage. The diffraction peaks at 2θ = 20.9° and 26.7° were the characteristic peaks of quartz impurity (JCPDS 01-0649), and the peak at 2θ = 31.0° was the characteristic peak of aragonite impurity (JCPDS 01-0628). The diffraction peaks at 2θ = 44.77°, 65.18° in the XRD patterns of nFe/Ni, ATP-nFe/Ni and D-ATP-nFe/Ni were assigned to the characteristic peaks of Fe^0^ (JCPDS 06-0696), thus indicating that Fe^0^ was successfully loaded on the attapulgite. However, the diffraction peaks of Ni were not observed in the XRD pattern due to the negligible content of Ni in the materials.

#### 3.1.3. Physicochemical Properties

The BET specific surface area, pore parameters and average particle size of different materials were measured and presented in Table 1. It is obvious that the physicochemical properties of ATP were greatly changed by the ball milling-freezing treatment, in which the specific surface area increased from 129.41 m^2^/g to 151.28 m^2^/g and the pore volume increased from 0.40 cm^3^/g to 0.52 cm^3^/g, whereas the average particle size decreased from 46.61 nm to 39.66 nm. This was mainly due to the synergistic effect of ball milling and freeze drying, which formed a fluffy structure on the attapulgite, thus increasing the disaggregation of ATP crystal bundles [29].

Furthermore, the specific surface area of nFe/Ni was 14.15 m^2^/g with a pore volume of 0.09 cm^3^/g. The specific surface area and pore volume of ATP-nFe/Ni increased significantly when nFe/Ni was loaded on ATP. The specific surface area and pore volume of D-ATP-nFe/Ni reached 97.10 m^2^/g and 0.32 cm^3^/g. Therefore, the specific surface area and pore volume of D-ATP-nFe/Ni were 6.9 times and 3.6 times greater than those of nFe/Ni, respectively. In terms of average particle size due to the clustering of nFe/Ni particles, the particle size of nFe/Ni cluster was as high as 423.94 nm, while the average particle size of D-ATP-nFe/Ni was 54.51 nm, thus indicating that the dispersion of D-ATP reduced the size of the nFe/Ni cluster, which was consistent with the TEM analysis.

### 3.2. Enhancement Effect of 2,4-DCP Dechlorination by nFe/Ni Supported on Disaggregated Attapulgite

Figure 3 shows the concentration changes of 2,4-DCP and the final product (phenol) in the process of removing 2,4-DCP by the different materials. The removal rate of 2,4-DCP by ATP and D-ATP within 120 min was less than 5% and the final product phenol was not observed, in which the removal of 2,4-DCP was caused by the adsorption of attapulgite. 2,4-DCP could be dechlorinated by nFe/Ni. The degradation rate of 2,4-DCP reached 78.5% and the yield of phenol was 53.9% within 60 min. After nFe/Ni was loaded on ATP and D-ATP, the degradation rate of 2,4-DCP and the yield of phenol increased diversely. The degradation rate of 2,4-DCP by ATP-nFe/Ni and D-ATP-nFe/Ni were 81.9% and 96.8% within 120 min, and the yields of phenol were 57.2% and 86.1%, respectively.

By comparing the apparent reaction rate constant *K*_obs_, it was found that the *K*_obs_ of the degradation of 2,4-DCP by D-ATP-nFe/Ni was 0.0277 min^−11^, while *K*_obs_ by nFe/Ni and ATP-nFe /Ni were 0.0125 min^−1^ and 0.0135 min^−1^, respectively. It was obvious that the reaction rate of D-ATP-nFe/Ni was much higher than that of ATP-nFe/Ni and nFe/Ni, indicating the nFe/Ni supported by disaggregated attapulgite showed a higher reactivity during the degradation of 2,4-DCP.

Compared with ATP and nFe/Ni, the degradation and reaction rates of removing 2,4-DCP by nFe/Ni loaded on D-ATP increased, which was mainly because D-ATP could disperse nFe/Ni nanoparticles and weaken the oxidation of Fe^0^. The effective iron percentage (Fe/Fe_T_) in nFe/Ni and D-ATP-nFe/Ni was measured and listed in Table 2. It could be observed that the percentage of Fe/Fe_T_ in nFe/Ni was only 51.5%, while that in D-ATP-nFe/Ni reached 97.8%. Although the percentage of Fe in nFe/Ni was higher than that in D-ATP-nFe/Ni, the content of reductive Fe in D-ATP-nFe/Ni was nearly 100%, indicating that only a small amount of Fe in D-ATP-nFe/Ni was oxidized due to the load of D-ATP.

The schematic representation of the disaggregation of ATP by the ball milling–freezing process and dispersion of nFe/Ni by D-ATP as a support is shown in Figure 4. After the disaggregation of the multiple stacking of ATP crystal bundles by the ball milling–freezing process, Fe/Ni nanoparticles could be blocked effectively by the steric hindrance of single ATP crystal rods, thus making it harder to aggregate together by magnetic attraction and nano effects. Single Fe/Ni spherical nanoparticles were formed after the chain-like structures were destroyed, thus providing more reaction active sites. In addition, the crisscrossing ATP crystal rods could provide a relatively anoxic environment for Fe/Ni nanoparticles, which reduced the contact oxidation of nFe with air. Compared with nFe/Ni and ATP-nFe/Ni, the disaggregated attapulgite could give full play to the reaction activity of nFe/Ni in the degradation of 2,4-DCP at the same concentration. Therefore, the degradation and reaction rates of 2,4-DCP were significantly improved.

### 3.3. Reaction Pathway and Kinetics of 2,4-DCP Dechlorination by nFe/Ni Supported on the Disaggregated Attapulgite

The concentration variations of 2,4-DCP and degradation products during the degradation of 2,4-DCP by D-ATP-nFe/Ni are shown in Figure 5a, where the degradation products were 2-CP, 4-CP and phenol. The concentration of 2,4-DCP in the solution gradually decreased to 69.1%, 38.1% and 6.8% at 20 min, 45 min and 90 min, respectively. The concentration of 2,4-DCP was only 3.1% within 120 min, which was almost completely removed. The concentration of the intermediate products 2-CP and 4-CP showed a trend of first increasing and then decreasing, before reaching the maximum value in 45 min. Then, the concentration gradually decreased due to the removal of another chlorine atom to generate phenol. Moreover, the production rate of Cl^−^ resembled that of phenol, where the concentration gradually increased with the extension of the reaction time. The yield of phenol reached 86.1% within 120 min.

Figure 5b gives a more explicit view of the mass balance analysis of the products during the degradation of 2,4-DCP. A total of 30.9% of 2,4-DCP was converted into 2-CP (6.8%), 4-CP (3.1%) and phenol (13.5%) within 20 min. At 60 min, the degradation rate of 2,4-DCP increased to 77.9%, while 9.5% 2-CP, 6.8% 4-CP and 54.6% phenol were generated. The degradation rate of 2,4-DCP reached 96.9% and the yield of phenol was 86.1% within 120 min eventually. Significantly, the yield of phenol did not reach 100% with a loss of mass of about 14%, which could be attributed to adsorption on the Fe^0^ conversion products [30,31,32].

Based on these analyses, the possible reaction pathways for 2,4-DCP dechlorination by D-ATP-nFe/Ni could be simplified to the sequence of steps, as shown in Figure 6, i.e., dichlorination by one step and dichlorination by two steps, which is described as: (1) two chlorine atoms were dissociated simultaneous from 2,4-DCP to generate phenol as final product, and (2) one chlorine atom was dissociated first with 2-CP or 4-CP as intermediate, and then another chlorine atom was removed to generate phenol.

According to the dechlorination degradation process of 2,4-DCP, the relationship between the concentration and time of each substance in the reaction process could be expressed as follows:(1)$$C_{2,4-\mathrm{DCP}\;}{=C}_{2,4-\mathrm{DCP},0}e^{-\left(k_{1}{+k}_{2}{+k}_{5}\right)t}$$
(2)$$C_{2-\mathrm{CP}\;}{=C}_{2,4-\mathrm{DCP},0}\;\times \;\left[\frac{k_{1}}{k_{3}\;-\;\left(k_{1}{+k}_{2}{+k}_{5}\right)}\left(e^{-\left(k_{1}{+k}_{2}{+k}_{5}\right)t}{-e}^{{-k}_{3}t}\right)\right]$$
(3)$$C_{4-\mathrm{CP}\;}{=C}_{2,4-\mathrm{DCP},0}\;\times \;\left[\frac{k_{2}}{k_{4}\;-\;\left(k_{1}{+k}_{2}{+k}_{5}\right)}\left(e^{-\left(k_{1}{+k}_{2}{+k}_{5}\right)t}{-e}^{{-k}_{4}t}\right)\right]$$
(4)$$C_{P}{\;=C}_{2,4-\mathrm{DCP},0}{\;-\;C}_{2-\mathrm{CP}}{\;-\;C}_{4-\mathrm{CP}}{\;-\;C}_{2,4-\mathrm{DCP}}$$
where *k*_1_~*k*_5_ (min^−1^) represent the step-by-step transformation rate of each organic. The values of *k*_1_~*k*_5_ were calculated by the non-linear least square fitting: *k*_1_ = 0.0070 min^−1^, *k*_2_ = 0.0052 min^−1^, *k*_3_ = 0.0253 min^−1^, *k*_4_ = 0.0390 min^−1^, and *k*_5_ = 0.0156 min^−1^. According to the reaction rates calculated above, the reaction rate for the direct dechlorination of 2,4-DCP of phenol (*k*_5_ = 0.0156 min^−1^) was much higher than those of 4-CP (*k*_2_ = 0.0052 min^−1^) and 2-CP (*k*_1_ = 0.0070 min^−1^), suggesting that the main dechlorination degradation pathway for the removal of 2,4-DCP by D-ATP-nFe/Ni was directly reduced to phenol by the simultaneous removal of two chlorine atoms, which was owed to the catalyst Ni in the reaction system that could catalyze the dissociation of H_2_ to active hydrogen atom (H*) by the reaction between Fe^0^ and water, thus improving the efficiency of dechlorination by breaking both C-Cl bond on the benzene ring simultaneously. In addition, one chlorine atom dissociated first with 2-CP or 4-CP as intermediate and then another chlorine atom was removed to generate phenol was the secondary pathway. The reaction rates *k*_1_ and *k*_2_ of 2-CP and 4-CP were far lower than the reaction rates *k*_3_ and *k*_4_ of the final product phenol. Because dechlorination is a reaction process initiated by adsorption, the dechlorination reaction rate gradually accelerated with the increasing adsorption capacity of the reaction, so the rate controlling step of the secondary pathway was one chlorine atom dissociated first with 2-CP or 4-CP as intermediate. Meanwhile, the reaction rate *k*_1_ for 2,4-DCP dechlorination to 2-CP was found to be faster than *k*_2_ to 4-CP, suggesting that the D-ATP-nFe/Ni was likely to generate 2-CP by removing the chlorine atom in the para-position of the benzene ring, which is consistent with the theory that the chlorine in the para-position of the phenolic hydroxyl group was more prone [33]. Hence, the main dechlorination degradation pathway of 2,4-DCP by D-ATP-nFe/Ni was directly reduced to phenol by the simultaneous removal of two chlorine atoms, and the secondary pathway was one chlorine atom dissociated first with 2-CP or 4-CP as intermediate and then another chlorine atom was removed to generate phenol.

## 4. Conclusions

The attapulgite crystal bundles could be disaggregated to single crystal rods with a length of 0.5–1.5 μm and a diameter of 20–40 nm by the synergistic effect of ball milling and freezing processes. The specific surface area and pore volume of D-ATP-nFe/Ni in which nFe/Ni was loaded on the disaggregated attapulgite reached 97.10 m^2^/g and 0.32 cm^3^/g, respectively. Fe/Ni nanoparticle clusters were dispersed into single spherical particles by ball milling–freezing disaggregated attapulgite, where the average particle size decreased from 423.94 nm to 54.51 nm. Therefore, the degradation rate of 2,4-DCP increased from 81.9% by ATP-nFe/Ni application to 96.8% by D-ATP-nFe/Ni application within 120 min, and the yield of phenol increased from 57.2% to 86.1%. Meanwhile, the reaction rate *K*_obs_ of 2,4-DCP degradation by D-ATP-nFe/Ni was 0.0277 min^−1^, which was higher than that of ATP-nFe/Ni (0.0135 min^−1^). The depolymerized attapulgite could disperse nFe/Ni nanoparticles and weaken the oxidation of Fe^0^, thus significantly improving the dechlorination degradation rate and reaction rate of 2,4-DCP. In the dechlorination process of 2,4-DCP by D-ATP-nFe/Ni, the reaction rate for the direct dechlorination of 2,4-DCP of phenol (*k*_5_ = 0.0156 min^−1^) was much higher than that of 4-CP (*k*_2_ = 0.0052 min^−1^) and 2-CP (*k*_1_ = 0.0070 min^−1^), thus suggesting that the main dechlorination degradation pathway for the removal of 2,4-DCP by D-ATP-nFe/Ni was directly reduced to phenol by the removal of two chlorine atoms. In the secondary pathway, the removal of one chlorine atom from 2,4-DCP to generate 2-CP or 4-CP as intermediate was the rate controlling step, and the final dechlorination product phenol was obtained as the dechlorination rate accelerating step with the progress of the reaction.

## Figures and Tables

**Figure 1 materials-15-03957-f001:**
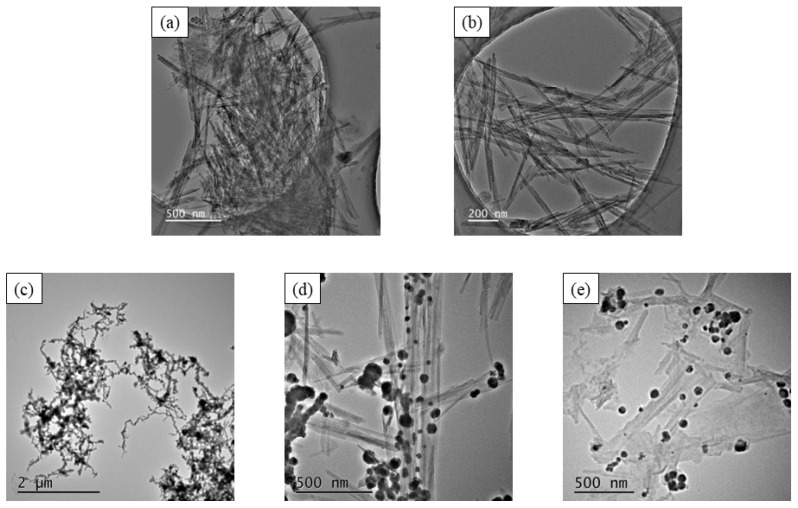
TEM images of (**a**) ATP, (**b**) D-ATP, (**c**) nFe/Ni, (**d**) ATP-nFe/Ni and (**e**) D-ATP-nFe/Ni.

**Figure 2 materials-15-03957-f002:**
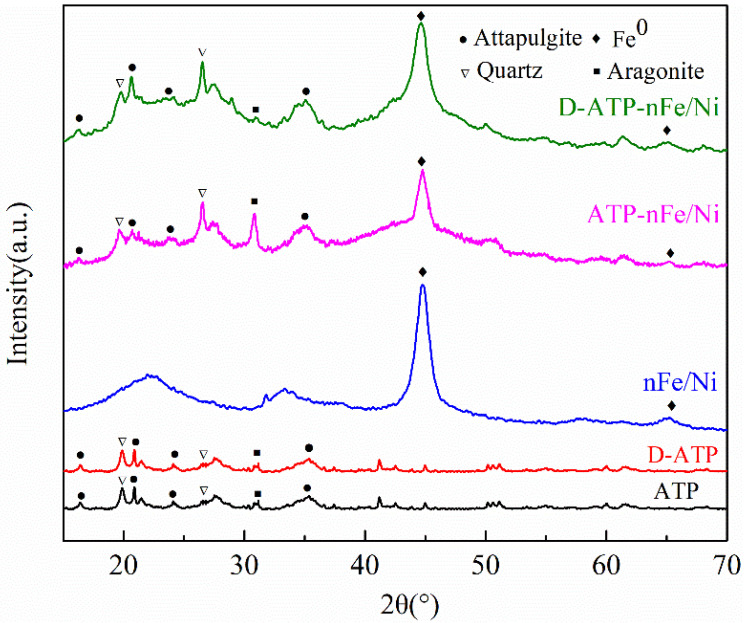
XRD patterns of ATP, D-ATP, nFe/Ni, ATP-nFe/Ni and D-ATP-nFe/Ni.

**Figure 3 materials-15-03957-f003:**
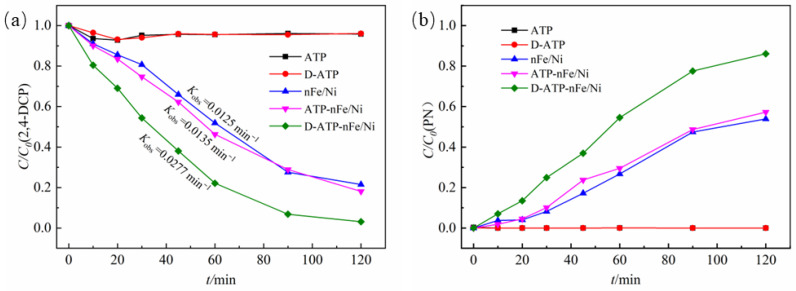
The concentration changes of (**a**) 2,4-DCP and (**b**) phenol.

**Figure 4 materials-15-03957-f004:**
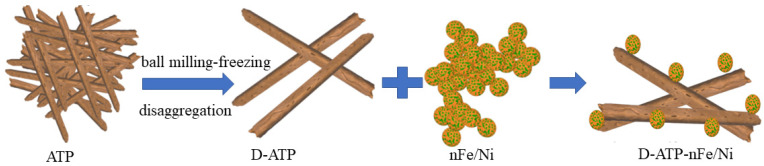
Schematic representation of the disaggregation of ATP by the ball milling–freezing process and dispersion of nFe/Ni by D-ATP as a support.

**Figure 5 materials-15-03957-f005:**
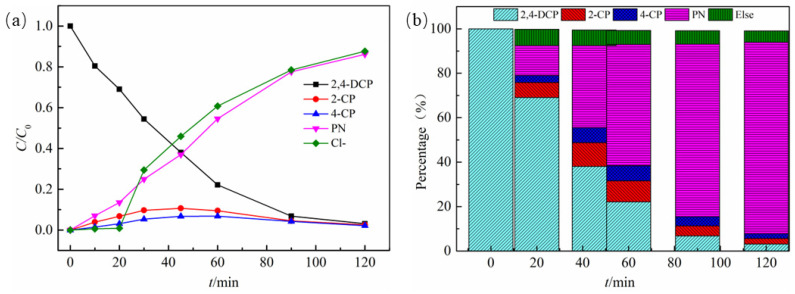
(**a**) Concentration variations of substance with time during the degradation of 2,4-DCP. (**b**) Mass balance analysis during the degradation of 2,4-DCP.

**Figure 6 materials-15-03957-f006:**
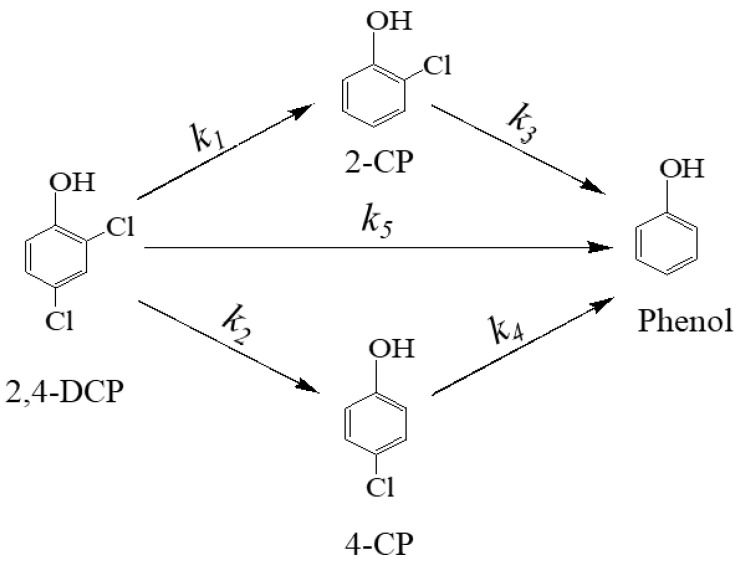
Reaction pathways for 2,4-DCP dichlorination by D-ATP-nFe/Ni.

**Table 1 materials-15-03957-t001:** Specific surface area, pore parameters and average size of materials.

Material	Surface Area /(m^2^·g^−1^)	Pore Volume /(cm^3^/g)	Nanoparticle Size (nm)
ATP	128.73	0.40	46.61
D-ATP	151.28	0.52	39.66
nFe/Ni	14.15	0.09	423.94
ATP-nFe/Ni	69.39	0.24	86.46
D-ATP-nFe/Ni	97.10	0.32	54.51

**Table 2 materials-15-03957-t002:** Percentage of Fe/Fe_T_ in nFe/Ni and D-ATP-nFe/Ni.

Material	* Fe_T_%	** Fe%	Fe/Fe_T_%
nFe/Ni	95.8	49.3	51.5
D-ATP-nFe/Ni	40.2	39.3	97.8

* Fe_T_% represents the total amount of iron contained in the material; ** Fe% represents the amount of Fe^0^ contained in the material.

## Data Availability

Not applicable.

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
