# Peer review of "Performance, Reaction Pathway and Kinetics of the Enhanced Dechlorination Degradation of 2,4-Dichlorophenol by Fe/Ni Nanoparticles Supported on Attapulgite Disaggregated by a Ball Milling–Freezing Process"

_materials, 2022, doi:10.3390/ma15113957_

Round 1
Reviewer 1 Report
The manuscripts shows that dispersing Fe and Ni nanoparticles in disaggregated attapulgite needles promotes de degradation of dichlorophenol. The manuscript can be published upon some modification that are described below:
-How can be proven that the Fe nanoparticles are not oxidized due to the support?
The authors should experimentally show that supporting Fe nanoparticles in attapulgite reduce oxidation of the nanoparticles.
-I suggest to revise the English in the manuscript thoroughly.
-The experimental section should be written in more detail.
For example, the catalytic experiments require hydrogen to peform the hydrodechlorination reaction, but there are no details in there, not about the use of a reductant (H2) or which are the conditions of the reductant.
Reviewer 2 Report
The manuscript is nicely written. Below are some minor comments:
- Are there any metal chlorides after treatment by ball milling?
- The selectivity toward chlorine removal in reference to the removal of the OH group requires further explanation.
- Would it be possible to do SEM-EDX analysis? This would enable to establish the presence of chlorine in the residue.
- The exact effect of operational parameters (such as rpm speed, ball sizes, Temperature) deserve more explanation.
- From a another important angle, self-condensation of phenols is well known to produce dioxins, thus, this could be mentioned in the introduction.
Reviewer 3 Report
The manuscript reported on the support of Fe/Ni bimetallic nanoparticles with attapulgite disaggregated by ball milling-freezing process. The new material was used for the dechlorination degradation of 2,4-dichlorophenol, which is a known pollutant in the environment, and is widely used in large scale production of many materials including pesticides, disinfectants, dyes and leather.
The abstract, introduction, and methodology captured the aim and objectives of the study adequately, and the results were well presented and discussed. However, my few observations are highlighted below.
- Although, chlorinated phenols are more toxic than the unsubstituted/unchlorinated one; and 96% degradation of 2,4-DCP is laudable. Notwithstanding, 86.1% generation of phenol as by-product seems to call for concern. If this is generated after wastewater treatment and released to the environment; is it not going to trigger another environmental problem, considering its own toxicity as well?
- Lines 97, 103, 122 - It is advisable to always start a sentence with word, and not figure, e.g. 5 g, 2.24 g and 0.4 g.
